# Spatio-Temporal Variation Characteristics of Snow Depth and Snow Cover Days over the Tibetan Plateau

**Chi Zhang [1], Naixia Mou [1,\*], Jiqiang Niu [2,3], Lingxian Zhang [1] and Feng Liu [1,\*]**

[1] College of Geodesy and Geomatics, Shandong University of Science and Technology, Qingdao 266590, China; 201883020096@sdust.edu.cn (C.Z.); 993851@sdust.edu.cn (L.Z.)

[2] School of Geographic Sciences, Xinyang Normal University, Xinyang 464099, China; niujiqiang@xynu.edu.cn

[3] Key Laboratory for Synergistic Prevention of Water and Soil Environmental Pollution, Xinyang Normal University, Xinyang 464099, China

[\*] Correspondence: mounx@lreis.ac.cn (N.M.); luf3286@sdust.edu.cn (F.L.)

**Abstract:** Changes in snow cover over the Tibetan Plateau (TP) have a significant impact on agriculture, hydrology, and ecological environment of surrounding areas. This study investigates the spatio-temporal pattern of snow depth (SD) and snow cover days (SCD), as well as the impact of temperature and precipitation on snow cover over TP from 1979 to 2018 by using the ERA5 reanalysis dataset, and uses the Mann–Kendall test for significance. The results indicate that (1) the average annual SD and SCD in the southern and western edge areas of TP are relatively high, reaching 10 cm and 120 d or more, respectively. (2) In the past 40 years, SD (s = 0.04 cm decade$^{-1}$, p = 0.81) and SCD (s = −2.3 d decade$^{-1}$, p = 0.10) over TP did not change significantly. (3) The positive feedback effect of precipitation is the main factor affecting SD, while the negative feedback effect of temperature is the main factor affecting SCD. This study improves the understanding of snow cover change and is conducive to the further study of climate change on TP.

**Keywords:** climate change; the Tibetan Plateau; snow depth; snow cover days; Mann–Kendall test

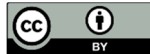

## 1. Introduction

The Tibetan Plateau (TP) is an important snow-covered area in the mid-latitudes of the northern hemisphere [1]. The snow cover over TP adjusts the surface energy balance through albedo feedback, thereby affecting the atmospheric circulation and climate system in East Asia [2–4]. Snowmelt can replenish water sources for river runoff over TP [5], effectively promote hydrological cycles [6,7], and thus affect agricultural production, hydropower generation, and change the ecological environment [8,9]. Therefore, studying the spatial and temporal characteristics of snow cover over TP is of important scientific significance.

In recent decades, global warming has accelerated. According to the Intergovernmental Panel on Climate Change [10], between 1951 and 2005 the average temperature in China increased by 1.3 °C, which has had an adverse impact on snow cover [11,12]. Many studies have analyzed the changing characteristics of snow depth (SD) and snow cover days (SCD), but views on snow cover changes are rather different. Qin and Liu [13], with Scanning Multichannel Microwave Radiometer (SMMR) dataset, identified that the annual cumulative daily snow depth of TP increased by 2.3 % year$^{-1}$ from 1957 to 1998. However, Ma et al. [14], with observations of meteorological stations, suggested that SD and SCD in most areas of TP decreased from 1971 to 2000. Based on Moderate Resolution Imaging Spectroradiometer (MODIS) dataset analysis, Sun et al. [15] and Wang et al. [16] found that the snow cover over TP has had a downward trend since 2000, while Duo et al. [17] and Wang et al. [18] found that the snow cover has not changed significantly since

2000. Bian et al. [19] analyzed the datasets of the National Aeronautical and Space Administration Modern-Era Retrospective Analysis for Research and Applications Version 2 (MERRA-2), the Japanese 55-year Reanalysis (JRA-55), and passive microwave (MW) satellite observations from 1980 to 2018, and found that the average annual SD shows a downward trend to varying degrees in the three datasets, while SCD presents a downward trend in the JRA-55 and MW datasets, and an upward trend in the MERRA-2 dataset. Therefore, it is necessary to further analyze the change of snow cover over TP in the past 40 years.

The variation of snow cover is affected by many factors, such as precipitation, temperature, and altitude, but the effect degree is not clear [20]. Li et al. [21] found that precipitation is the main factor affecting the change of snow cover over TP from 2001 to 2014, however, Li [22] and Hu and Liang [23] noted that SD of TP is positively correlated with the average temperature of the northern hemisphere, and that temperature changes have the most obvious effect on increasing snowmelt. Some studies have also demonstrated that the competing effects of temperature and precipitation simultaneously affect snow cover to different degrees. For example, Ke et al. [24] believe that snow cover is negatively correlated with winter temperature, and positively correlated with precipitation in winter and spring. Wang et al. [25] emphasized that temperature is a key factor affecting snow cover in autumn and spring, while precipitation is a key factor in winter. Therefore, it is necessary to analyze the relationship between TP snow cover and climatic factors (temperature and precipitation). In addition, some studies have pointed out that the warming of high-altitude areas is stronger than that of low-altitude areas, which leads to the difference in the distribution and change of snow cover in different altitude areas [26–28]. However, few studies have investigated the impacts of climatic factors on snow cover variation in different altitudes of TP, which is made up for by this paper.

A series of studies have been conducted to investigate the distribution and change of snow cover by using in situ observations and remote sensing products [29–32]. Since most of the meteorological stations of the China Meteorological Administration (CMA) are located in the eastern part of TP, with few stations in the high-altitude areas of the central and western regions, the representativeness of using local-scale observation for TP as a whole is questionable [33]. Remote sensing data are widely used in optical remote sensing (MODIS), passive microwave remote sensing (the Advanced Microwave Scanning Radiometer for EOS(AMSR-E), the Special Sensor Microwave Image(SSM/I), etc.) [34–37]. MODIS has a high spectral and spatial resolution. However, MODIS' short record (2000–present) obscures understanding of snow's long-term response to climate change [38–40]; passive microwave remote sensing products are not affected by weather, but the monitoring of scattered snow in mountainous areas is prone to deviations due to their low spatial resolution [30,41]. In recent years, the rapid development of the reanalysis dataset has made a compromise between station data and satellite data [42–45]. A reanalysis dataset is obtained through data assimilation models (assimilating satellite, surface, and upper-air conventional observation data) and surface process model simulations [46,47]. With advantages of long time series and wide monitoring range, it is suitable for analysis of the long-term variation trend of snow cover and the information of cryosphere climate feedback, and so forth. [48,49]. It should be noted that under the influence of assimilation data, the reanalysis data has strong inhomogeneities, so there is certain uncertainty in the trend analysis.

Based on the ERA5 dataset including SD, temperature, and precipitation provided by the European Centre for Medium-Range Weather Forecast (ECMWF), a long-term and large-scale spatio-temporal characteristic analysis of SD and SCD from 1979 to 2018 is conducted. More specifically, (1) a comparative analysis of the differences in spatio-temporal patterns of SD and SCD over TP is considered; (2) the inter-annual variations of SD and SCD are quantified, and their changing trends at different altitudes are analyzed; (3) the relationships between climatic factors (precipitation and temperature), altitudes, and SD and SCD are studied. The objective of this paper is to investigate the dynamic changes

of snow cover on TP and provide an effective reference for local water resource management, climate change research, and future snow cover prediction.

## 2. Materials and Methods

### 2.1. Study Area

TP, the highest and largest plateau in the world, is located in southwestern China and central Eurasia, with an average elevation of more than 4000 m and an area of more than $200 \times 10^4$ km² [50]. The range of this area in China is 26°00′12″ N to 39°46′50″ N and 73°18′52″ E to 104°46′59″ E, spanning all of Tibet and parts of Qinghai, Xinjiang, Gansu, Sichuan, and Yunnan. It is connected with the Kunlun Mountains, Altun Mountains, Qaidam Basin, and Qilian mountains in the north, including Karakoram Mountains and Pamir regions in the northwest. In the central region are Northern TP, the Tangula Mountains, Bayan Har Mountains, and Nyainqentanglha Mountains. The Himalayas are in the south and extend to the Hengduan Mountains in the southeast (Figure 1). As the water tower of Asia, TP is the source of many rivers (such as the Indus, Yangtze, Yellow, and Brahmaputra), providing fresh water for billions of people [51]. Snowfall is the surface characteristic of TP with the greatest seasonal variations. It is closely related to the East Asian and South Asian monsoons and impacts both drought and floods, including on the middle and lower reaches of the Yangtze River [52–54].

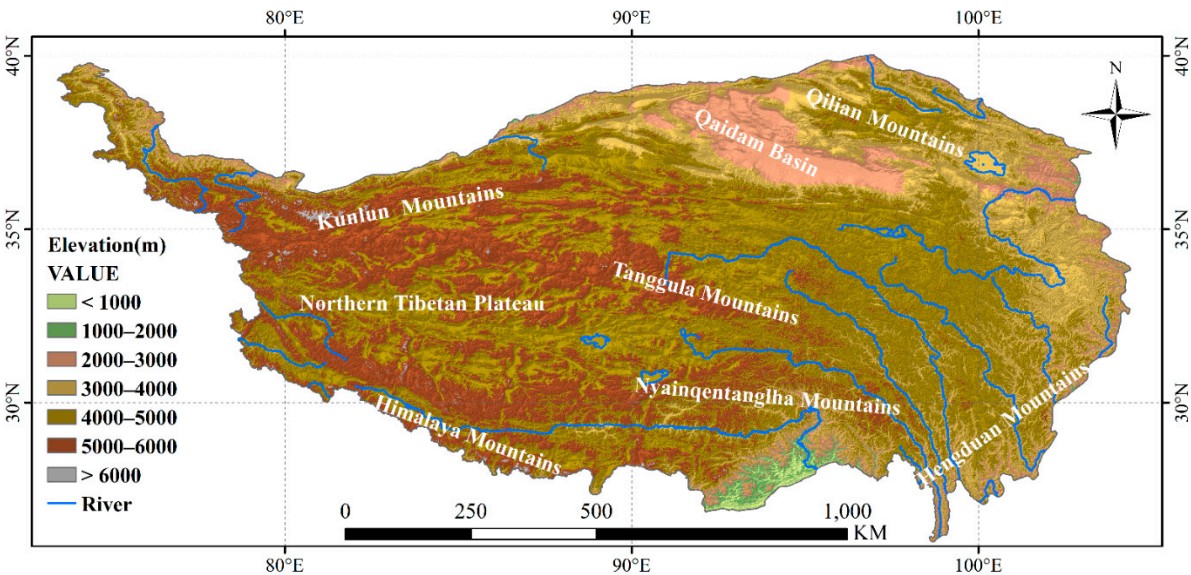

**Figure 1.** Physiographical regions and distribution of meteorological stations over the Tibetan Plateau (TP).

### 2.2. Data

#### 2.2.1. Snow Data

In this paper, ERA5 daily SD data in 0.25° × 0.25° grid from 1979 to 2018 were used to study the change of snow cover. ERA5 is produced by Copernicus Climate Change Service (C3S), and it is obtained by using extensive station and satellite measurement data modeling (land model version is HTESSEL, atmospheric model is the Integrated Forecast System (IFS) Cy41r2) and data assimilation analysis (4D-variational algorithm) (as detailed by Rosnay et al. [55] and Hersbach et al. [56]). IFS uses the surface data assimilation model to conduct two-dimensional optimal interpolation analysis of IMS(Interactive Multisensor Snow and Ice Mapping) snow cover data and station data, where the IMS dataset integrates various satellite images (Advanced Very High Resolution Radiometer(AVHRR), Geostationary Operational Environmental Satellites(GOES), etc.) and de-

rived mapping products (AMSR-E, National Centers for Environmental Prediction(NCEP) Model, etc.). National Oceanic and Atmospheric Administration(NOAA) has been in operation since 1997 as one of the assimilation data sources for IMS dataset. It has a nominal resolution of 24 km at the daily time scale. Since 23 February, 2004, the IMS has been distributed at a pixel size of 4 km. Since 2 December, 2014, the IMS has been distributed at a pixel size of 1 km. More information is provided by the official documentation, please see https://confluence.ecmwf.int/display/CKB/ERA5%3A+data+documentation. The dataset is available on the website https://cds.climate.copernicus.eu/.

Wang et al. [57], Liu et al. [58], Hersbach et al. [59], Terzago et al. [60], and Orsolini et al. [61] conducted a comprehensive assessment of snow cover parameters for this dataset, the findings showed that the ERA5 dataset can well capture the spatial distribution of snow cover and a wide range of characteristics of seasonal changes despite its large positive bias. Wang et al. [62], Matveeva and Sidorchuk [63], and Yılmaz et al. [64] also used this data for snow study.

The formula for the snow model in ERA5 dataset is as follows:

$$\text{snow cover}\,(\text{SC}) = \min\left(1, \left(\text{RW} \cdot \text{SD}/\text{RSN}\right)/0.1\right) \tag{1}$$

where RW is water density, RSN is snow density, SD is snow depth, and SC means snow cover. Equation (1) is given by https://confluence.ecmwf.int/display/CKB/ERA5%3A+data+documentation; it briefly describes the treatment of snow cover. More detail can be seen on the website https://www.ecmwf.int/sites/default/files/elibrary/2016/17117-part-iv-physical-processes.pdf#section.H.4.

SCD is obtained by accumulating the number of days with daily snow depth >1 cm.

$$\text{SCD} = \sum_{i=1}^{n} s_i \tag{2}$$

SCD represents snow cover days, which is the duration of snow cover, $s_i$ is a 0–1 variable, $s_i = 1$ and $s_i = 0$ respectively represent snow and no snow conditions.

SD and SCD are integrated into snow cover year (from 1 August to 31 July of next year). The snow cover year is divided into four seasons: spring (March to May), summer (June to August), autumn (September to November), and winter (December to February). It should be noted that snowfall mainly occurs in the cold season when temperature and precipitation have a greater impact on snow cover. Therefore, referring to the research of [65,66], this paper merely analyzes the spatio-temporal variation in autumn, winter, spring, and snow cover year together with their correlation with temperature and precipitation.

### 2.2.2. DEM Data

SRTM 90 DEM (V004) data were obtained from the National Map Seamless Data Distribution Systems (http://seamless.usgs.gov) with a spatial resolution of 90°. Nearest-neighbor interpolation was used to resample the original DEM data, and the spatial resolution was adjusted to 0.25° × 0.25°. DEM data were used to investigate the change of snow cover at different altitude intervals.

### 2.2.3. Climate Data

In order to study the relationship between snow cover and climatic factors (air temperature and precipitation), the ERA5 dataset was used to analyze the total precipitation and temperature of 2 m on the surface. He et al. [67], Xue et al. [68], Li et al. [69], and Wang et al. [70] verified the feasibility of the ERA5 dataset in analyzing climate change on TP. In addition, Josey et al. [71], Watterson [72], Tarek et al. [73], and Tang et al. [74] also used ERA5 temperature and precipitation datasets for hydrology modeling, climate change, and other studies.

### 2.3. *Theory and Method*

#### 2.3.1. Theil–Sen Slope

The Theil–Sen slope estimator (median trend) was used to analyze the trend of SD and SCD [75]. The calculation formula is as follows:

$$\beta = \text{Median}\left(\frac{x_i - x_j}{i - j}\right), \, i > j \tag{3}$$

where $i$ is a time series, $x_i$ is snow depth or snow days, $\beta > 0$ represents trend increase, and $\beta < 0$ represents trend decrease.

#### 2.3.2. Pearson Correlation Coefficients

Pearson correlation coefficients were used to analyze the relationship between SD/SCD and temperature/precipitation. Pearson correlation coefficients can measure the linear correlation between two variables $x$ and $y$. The calculation formula is as follows:

$$r = \frac{\sum_{i=1}^{n}\left(x_i - \bar{x}\right)\left(y_i - \bar{y}\right)}{\sqrt{\sum_{i=1}^{n}\left(x_i - \bar{x}\right)^2 \cdot \sum_{i=1}^{n}\left(y_i - \bar{y}\right)^2}} \tag{4}$$

where $x$ is the variable of SD or SCD, and $y$ represents temperature or precipitation. $r > 0$, $r < 0$, $r = 0$ respectively represent positive correlation, negative correlation, and no correlation between the two factors.

#### 2.3.3. Mann–Kendall

A Mann–Kendall method was used to test the significance level of SD and SCD. The Mann–Kendall method is a nonparametric statistical test method. When applied to the test of monotone nonlinear data, it will not be adversely affected by outliers [76]. In this study, the confidence level of 95% was used to evaluate the significance of trend and correlation. The calculation formula is as follows:

$$S = \sum_{i=1}^{n}\sum_{j=i+1}^{n}\text{sgn}\left(x_j - x_i\right) \tag{5}$$

$$\text{sgn}\left(x_j - x_i\right) = \begin{cases} +1, \text{ if } \left(x_j - x_i\right) > 0 \\ 0, \text{ if } \left(x_j - x_i\right) = 0 \\ -1, \text{ if } \left(x_j - x_i\right) < 0 \end{cases} \tag{6}$$

where $n$ is the number of years, $S > 0$ is an increasing trend, $S = 0$ represents no change, and $S < 0$ is a downward trend. If $n < 0$, $S$ can be directly used for bilateral trend test. Under a given significance level $\alpha$, if $|S| \leq S_\alpha/2$, the trend is significant; otherwise, it is not significant.

## 3. Results

### 3.1. *Spatial Distribution of SD and SCD*

Figure 2 illustrates the spatial distribution of average SD and SCD over TP in autumn, winter, spring, and snow cover years from 1979 to 2018. The spatial distribution patterns of SD and SCD are similar, both showing extremely high spatial heterogeneity. Overall, in snow cover years, areas with high SD (>5 cm) over TP account for 7%, and areas with high SCD (>240 d) account for 6%, mainly distributed in the Karakoram Mountains and the Nyainqentanglha Mountains, the southern edge of the Himalayas and the Qilian Mountains. Areas with low SD (<5 cm) account for 21%, and areas with low SCD (<20 d) account for 26%, mainly distributed in the valleys of southern Tibet, Yarlung Zangbo

River Valley, Northern Tibet Plateau, and Qaidam Basin. In winter, areas with SD > 1 cm and SCD > 60 d are the highest, accounting for 54% and 38%, respectively. In autumn, the areas with SD < 0.5 cm and SCD < 60 d are the highest, both accounting for 50%.

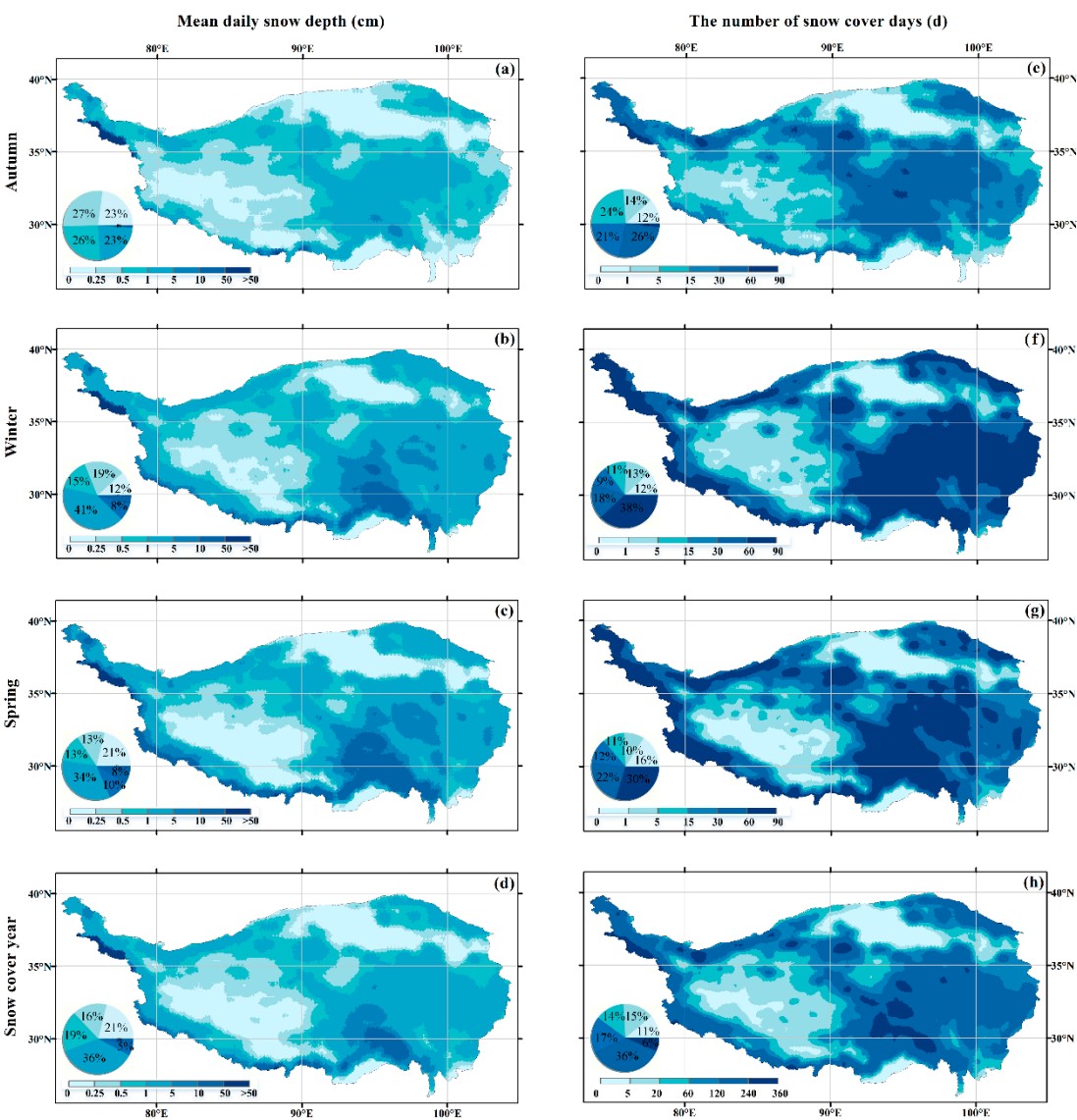

**Figure 2.** Spatial patterns of snow depth (SD) and snow cover days (SCD) from 1979 to 2018. Illustration in the lower left corner shows the frequency distribution of the mean. (**a–d**) are spatial distribution of SD in autumn, winter, spring and snow cover year. (**e–h**) are spatial distribution of SCD in autumn, winter, spring and snow cover year.

### 3.2. Spatial and Temporal Trends of SD and SCD

#### 3.2.1. Spatial Variation Trends of SD and SCD

Figure 3 shows the spatial variation trends of SD and SCD over TP from 1979 to 2018. In snow cover years, the areas with reduced SD and SCD account for 83% and 81%, respectively, among which the most obvious reduction trends are −5 cm decade⁻¹ and −1 d decade⁻¹ in Nyainqentanglha Mountain. However, in the Kunlun Mountains, Bayankera Mountains, and the southern edge of the Himalayas, SD and SCD both show increasing trends, 3 cm decade⁻¹ and 1 d decade⁻¹, respectively. It is worth noting that the Himalayas on the southern edge of the Tibetan Plateau are the areas where the freezing line rises [77]. In the central part of the Himalayas, SD and SCD show a decreasing trend, while in the northern and southern parts of the Himalayas, they show an increasing trend. Similarly, in the central and southern part of the Kunlun Mountains, there is an increasing trend,

while in the northern part of Kunlun Mountains, there is a decreasing trend. This may be due to the large area of the two mountains, the wide transverse span, the complex terrain, and so forth. The range of SD decrease in spring is the largest (80%), and the range of SD increase in winter is the largest (33%) In each season, SCD decreases across ~90% of the TP and increases across ~10% of the TP.

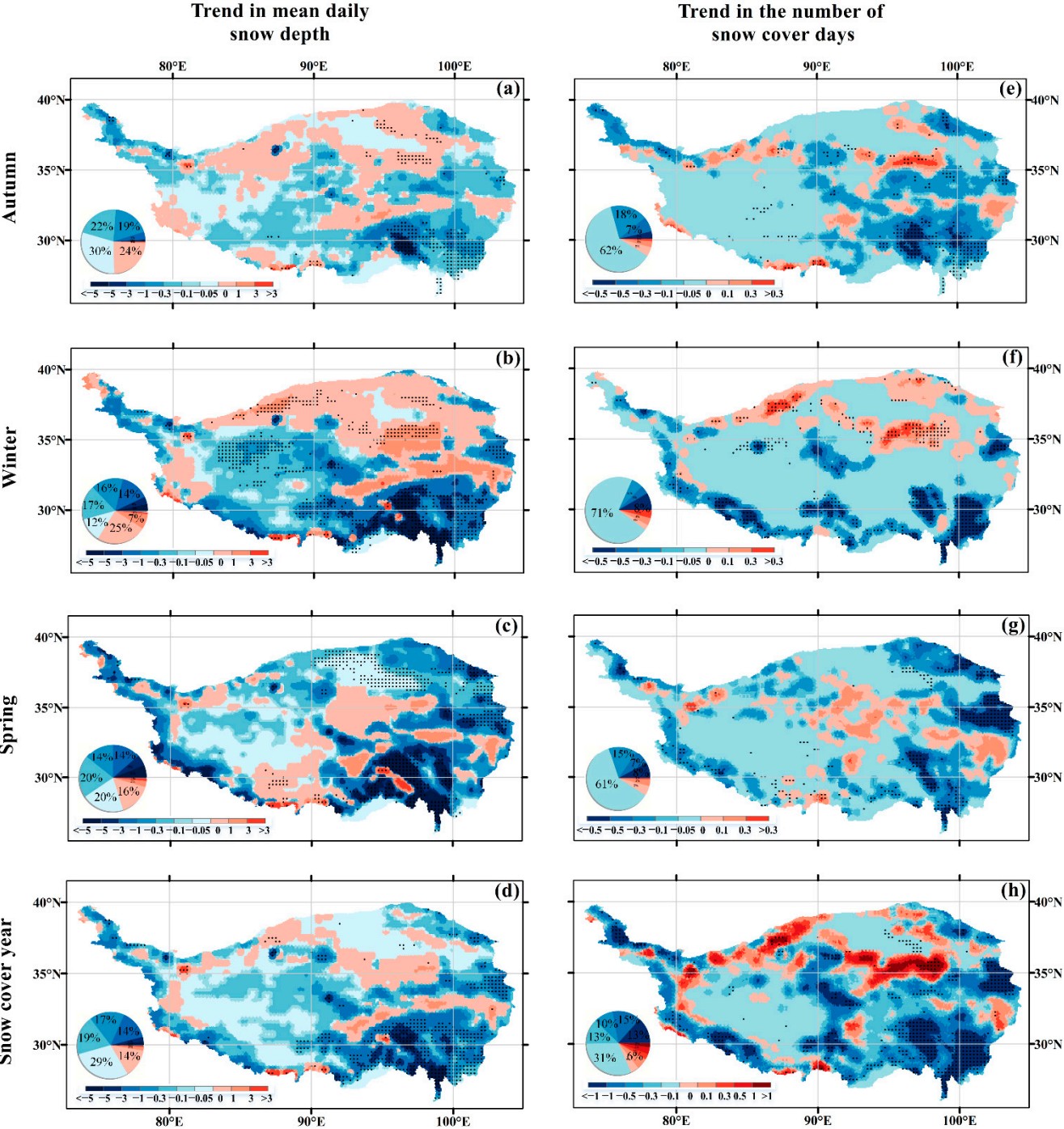

**Figure 3.** Trends in SD and SCD from 1979 to 2018. Illustration in the lower left corner of each sub-figure shows the frequency distribution of the trends, and the black dots indicate the areas that pass the significance test ($p < 0.05$). (**a**–**d**) are variation trends of SD in autumn, winter, spring and snow cover year. (**e**–**h**) are variation trends of SCD in autumn, winter, spring and snow cover year.

### 3.2.2. Interannual Variation Trends of SD and SCD

Figure 4 explores the interannual variation of snow cover over TP. The study found no significant change in SD (s = 0.04 cm decade$^{-1}$, $R^2$ = 0.0014, p > 0.05) or SCD (s = −2.33 d decade$^{-1}$, $R^2$ = 0.066, p > 0.05) from 1979 to 2018. It can be seen from Figure 4 that there was a slight increase in SD from 1979 to 2018, which may be related to the high value of SD in 2015–2018. Deeper snow raises the SD level throughout the cycle to a certain extent. Contrary to SD, SCD decreased slightly, with high values mainly concentrated before 2000, such as in 1983 (129 d), 1990 (121 d), and 1997 (130 d). Please see Figure S1 and Table S1 for interannual variations in different seasons.

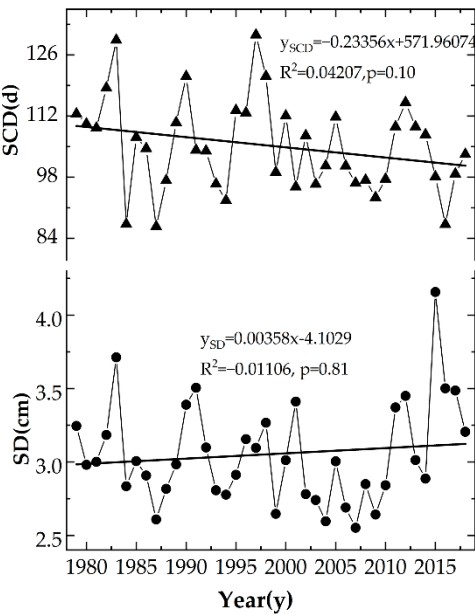

**Figure 4.** Interannual variability and trends in SD and SCD from 1979 to 2018.

### 3.2.3. Variation Trend of SD and SCD at Different Altitude Intervals

Figure 5 and Table 1 show the changes of SD and SCD at different altitude intervals. The study area was divided into equal altitude intervals with a constant altitude distance of 1000 m, and the interannual changes of SD and SCD in different altitude intervals over TP from 1979 to 2018 were analyzed. The results showed that SD and SCD both increased as the altitude increased. To be specific, when the altitude exceeds 6000 m, the annual average SD and SCD can reach more than 100 cm and 150 d, respectively. At the altitude of 5000–6000 m, SD increased significantly (s = 0.7 cm decade$^{-1}$, p = 0.02), which was caused by the sudden increase of SD at this altitude interval from 2015 to 2018. Moreover, the variation trend of SCD fluctuated greatly, and the overall average SCD was decreasing (s = −2.3 d decade$^{-1}$, p = 0.1). Please see Figures S2–S4 and Table S2–S4 for interannual variations of SD and SCD in different seasons and different altitudes. In addition, due to the significant regional differences between the Himalayas and Kunlun Mountains, we analyzed the interannual variation of snow cover at different altitudes in these two areas. Please see Figures S5 and S6 and Tables S5 and S6 of the Supplementary Files.

**Table 1.** Assimilation data source.

| Dataset Name | Observation | Measurement |
|---|---|---|
| SYNOP | Land station | Snow depth |
| Additional national reports | Land station | Snow depth |
| NOAA IMS | Merged satellite | Snow cover (NH only) |

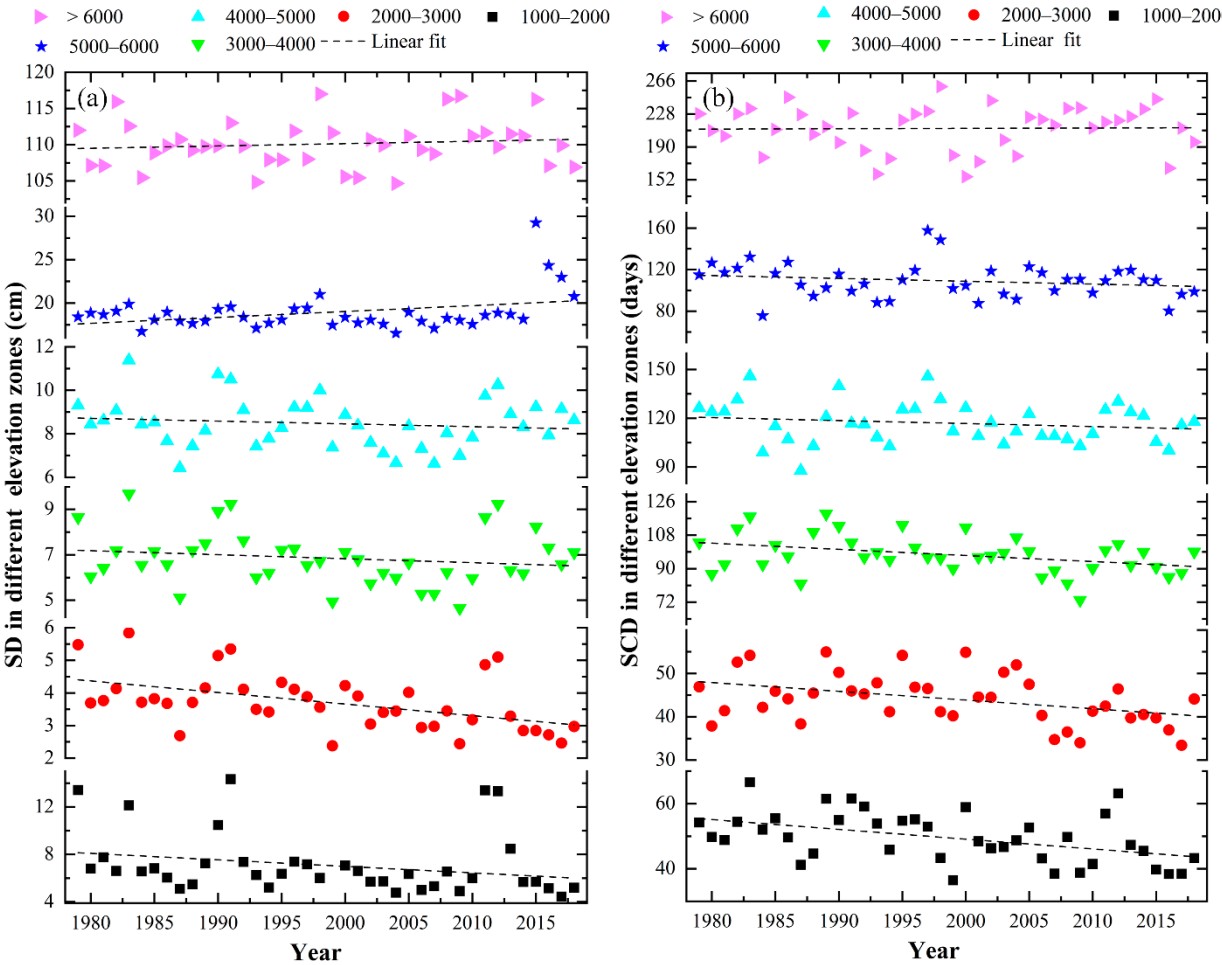

**Figure 5.** Interannual variability in the SD (**a**) and SCD (**b**) at various altitude intervals across TP during 1979–2018.

### 3.3. Correlation between SD/SCD and Temperature/Precipitation

3.3.1. Correlation between SD and Temperature/Precipitation

To analyze the climatic factors that affect snow variation, we used Pearson correlation coefficients to analyze the correlation between SD and temperature/precipitation (Figure 6). The findings showed that SD was negatively correlated with temperature in most parts of TP, while SD was highly heterogeneous with precipitation. In snow cover years, SD in 99% of the areas showed a negative correlation with temperature, of which 73% have a significant correlation. There is a negative correlation between SD and precipitation in 52% of the areas, mainly located in the Kunlun Mountains, southern Tibet, and the central and eastern parts of TP, and a positive correlation between SD and precipitation in 48% of the areas, mainly located in the northern TP, Qaidam Basin, and the southeastern edge of TP. Areas with negative correlation between SD and temperature ($-1 < r < -0.6$) concentrate in autumn and spring, accounting for 76% and 70%, respectively, while those with positive correlation ($0.4 < r < 1.0$) are most found in winter, accounting for 80%.

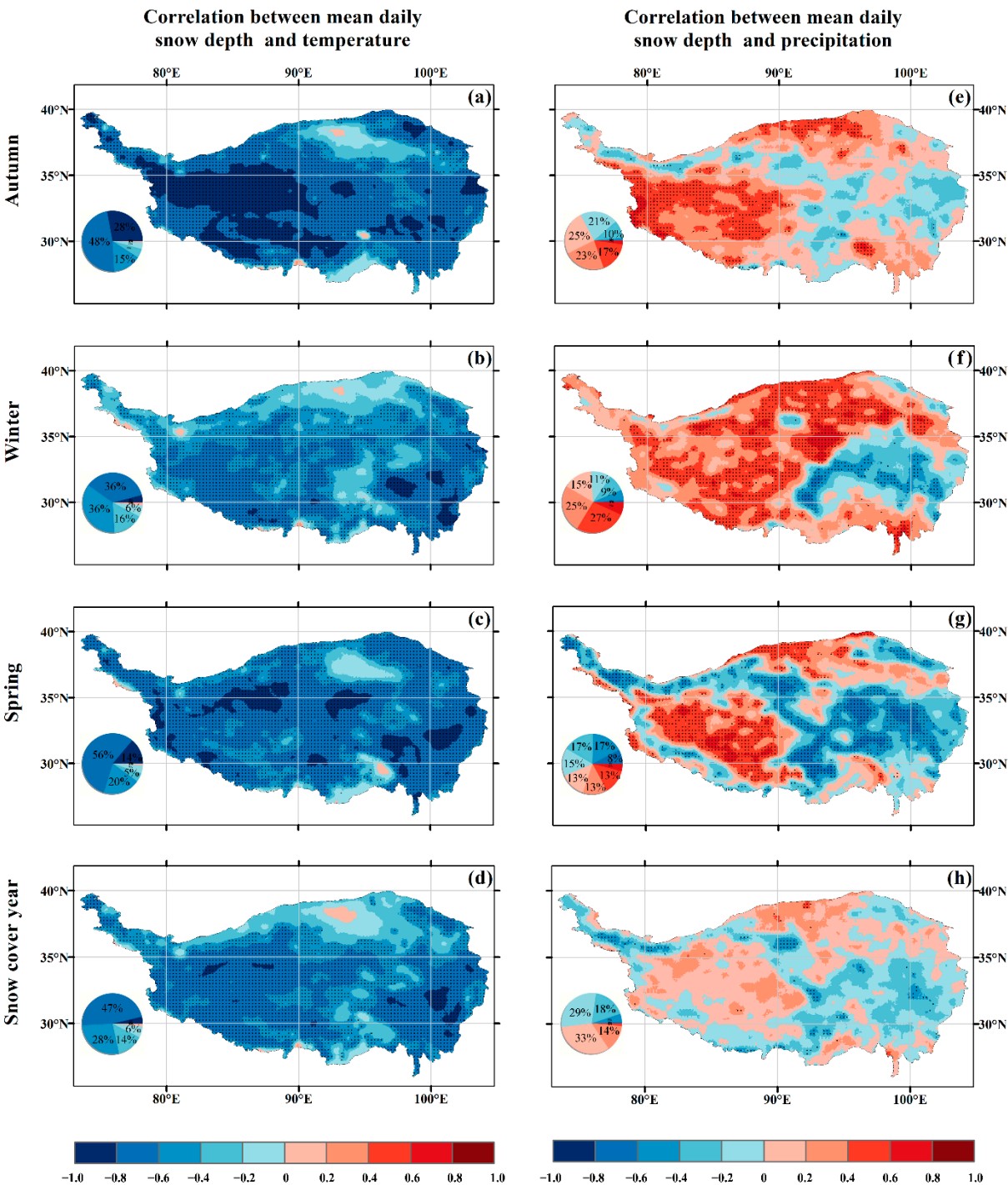

**Figure 6.** Correlation between mean snow depth and temperature/precipitation during 1979–2018. Illustration in the lower left corner of each sub-figure shows the frequency distribution of the correlation coefficient, and the black dots indicate areas that pass the significance test ( $p < 0.05$ ).(**a–d**) are correlation coefficients between SD and temperature in autumn, winter, spring and snow cover year. (**e–h**) are correlation coefficients between SD and precipitation in autumn, winter, spring and snow cover year.

### 3.3.2. Correlation between SCD and Temperature/Precipitation

Figure 7 analyzes the correlation between SCD and temperature/precipitation, which presents a spatial distribution characteristic similar to the correlation between SD and temperature/precipitation. Precisely, the correlation coefficient of the former is smaller than that of the latter. For example, in the northern TP, the negative correlation coefficient

between SCD and temperature is $-0.8 < r < -0.6$ in spring, and the positive correlation coefficient between SCD and the precipitation is $0 < r < 0.4$ in autumn.

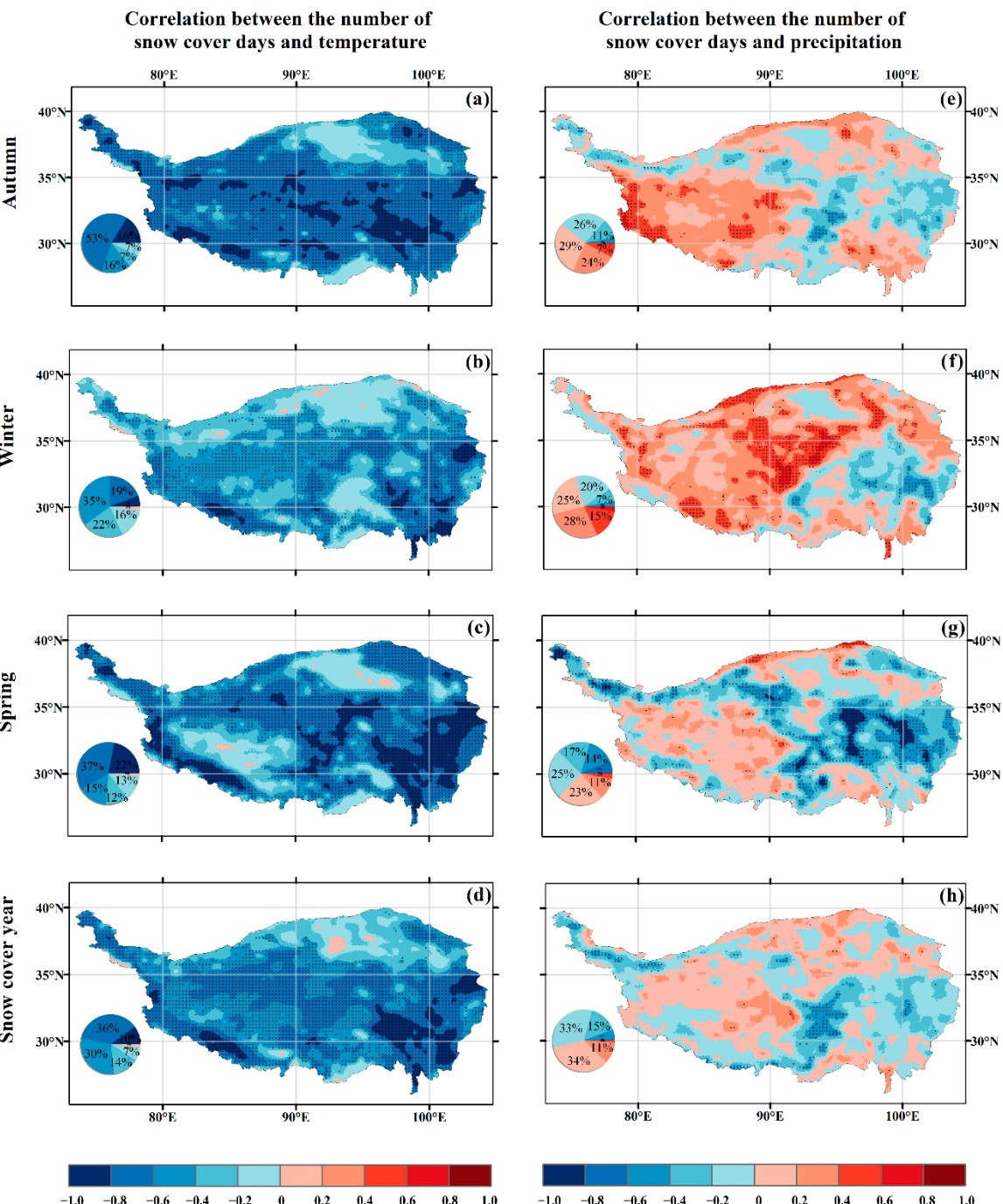

**Figure 7.** Correlation between SCD and temperature/precipitation during 1979–2018. Illustration in the lower left corner of each sub-figure shows the frequency distribution of the correlation coefficients, and the black dots indicate areas that pass the significance test ($p < 0.05$). (**a–d**) are correlation coefficients between SCD and temperature in autumn, winter, spring and snow cover year. (**e–h**) are correlation coefficients between SCD and precipitation in autumn, winter, spring and snow cover year.

### 3.3.3. Correlations between SD/SCD and Temperature/Precipitation at Different Altitude intervals

To further understand the spatial heterogeneity of the correlation between SD/SCD and temperature/precipitation, Figure 8 analyzes the variations of SCD and temperature

($R_{sd\text{-}t}$), SD and precipitation ($R_{sd\text{-}p}$), SCD and temperature ($R_{scd\text{-}t}$), and SCD and precipitation ($R_{scd\text{-}p}$) at different altitudes. The results show that as the altitude increases, the negative correlation of SD and SCD with temperature gradually increases, while the positive correlation with precipitation gradually decreases. $R_{sd\text{-}p}$ shows a significant positive correlation between SD and precipitation at 2500–5000 m. $R_{scd\text{-}t}$ shows a negative correlation between SCD and temperature, increasing gradually from 2500 m to 3500 m, then stabilizing at 3500–4700 m, and gradually decreasing over 4700 m. $R_{scd\text{-}p}$ indicates that SCD is positively correlated with precipitation, which decreases gradually from 2000 m to 4500 m, and increases slightly from 4500 m to 5500 m. In addition, we analyzed the correlation between the SD/SCD and temperature/precipitation of the Kunlun Mountains and the Himalayas at different altitudes. Please see Figures S7 and S8 of the Supplementary Files.

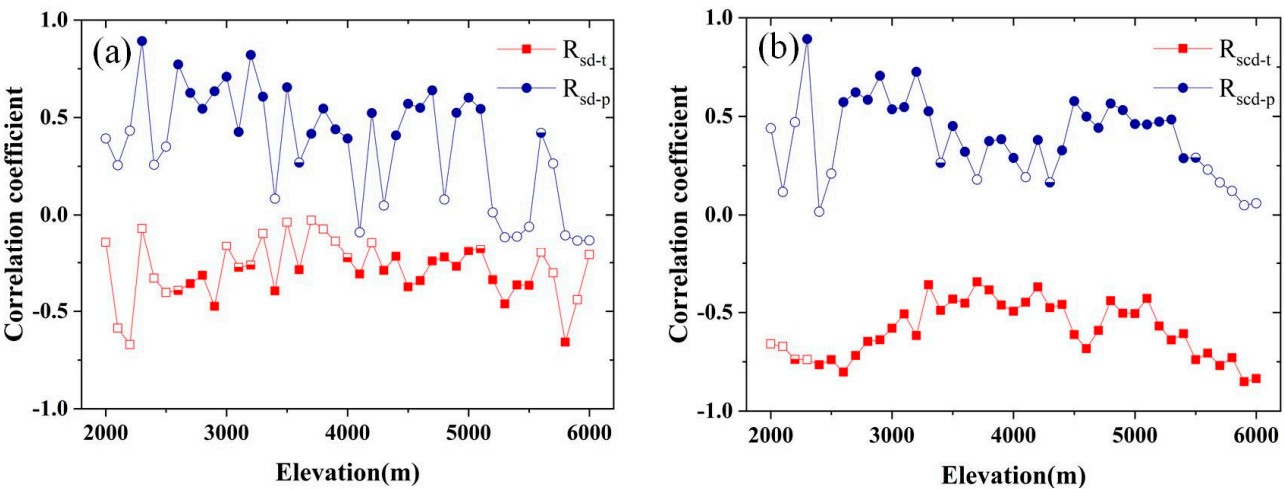

**Figure 8.** The correlation between SD (**a**)/SCD (**b**) and temperature/precipitation at different altitude intervals. The red line represents the correlation coefficient between SD/SCD and temperature, and the blue line represents the correlation coefficient between SD/SCD and precipitation. The solid points are significant at 99%, the half-filled points are significant at 95%, and the hollow points are insignificant.

The following analyses were conducted from the overall perspective, as is shown in Table 2. Pearson correlation coefficients between SCD and temperature in autumn, winter, spring, and snow cover years are −0.802, −0.704, −0.800, and −0.744 ($p < 0.01$), respectively. Pearson correlation coefficients between SD and temperature in winter and spring are −0.373 and −0.513 ($p < 0.01$), respectively. However, the correlations between SD/SCD and precipitation are statistically insignificant ($p > 0.05$).

**Table 2.** Pearson correlation coefficients between SD/SCD and temperature/precipitation of autumn, winter, spring, and snow cover year in the TP from 1979 to 2018. One and two asterisks denote significance at the 0.05 and 0.01 levels, respectively.

|  | Temperature | Precipitation | Temperature | Precipitation |
|---|---|---|---|---|
|  | **Autumn** | | **Winter** | |
| SCD | −0.802 ** | 0.101 | −0.704 ** | 0.301 |
| SD | −0.086 | 0.042 | −0.373 | 0.054 |
|  | Temperature | Precipitation | Temperature | Precipitation |
|  | **Spring** | | **Snow Cover Years** | |
| SCD | −0.800 ** | −0.289 | −0.744 ** | −0.107 |
| SD | −0.513 * | −0.160 | −0.053 | −0.210 |

## 4. Discussion

### 4.1. Analysis of the Causes of Spatio-Temporal Variation of Snow Cover

It was found that SD and SCD did not change significantly from 1979 to 2018, which is consistent with the researches of Wang et al. [18] and Xu et al. [78]. Bian et al. [19] analyzed the variation trend of the annual mean SD over TP from 1980 to 2018, and found that SD showed different decreasing trends for MERRA-2 (−0.03 cm decade$^{-1}$, p = 0.47), JRA-55 (−0.27 cm decade$^{-1}$, p = 0.00) and MW (−0.18 cm decade$^{-1}$, p = 0.00), while analyzing the variation trend of annual SCD, and found that SCD showed an insignificant increasing trend for MERRA-2 (0.38 d decade$^{-1}$, p = 0.86), and a significant decreasing trend for JRA-55 (−6.04 d decade$^{-1}$, p = 0.00) and MW (−3.88 d decade$^{-1}$, p = 0.01). This difference in trend has a certain relationship with the uncertainty of the reanalyzed data. (Details will be discussed in Section 4.3) Differently, Bian et al. focused on the whole TP, while this paper focuses instead on the part of TP in China. The area studied in this paper does not include parts of the Pamir Plateau and the Himalayas, which have more snow cover and are more variable.

We believe that the high spatial heterogeneity of TP snow cover may be one of the reasons for the insignificant changes in SD and SCD (Figure 3). Among them, the areas with a significant decrease in SD are mainly concentrated in the southwestern and a small part of the western of TP, among them, the Nianqing Tanggula Mountains has the most significant decrease (<−5 cm decade$^{-1}$, p < 0.05). The areas showing an increasing trend are mainly concentrated in the Kunlun Mountains, the Bayankala Mountains, and the southern edge of the Himalayas (>3 cm decade$^{-1}$, p > 0.05). Among them, over the Himalayas, SD shows significant increasing trends at 4500–5000 m (s = 0.02 cm decade$^{-1}$, p = 0.02) and 5000–5500 m (s = 0.09 cm decade$^{-1}$, p < 0.01), and insignificant increasing trends at 5500–6000 m (s = 0.2 cm decade$^{-1}$, p < 0.01). In the Kunlun Mountains, there is an insignificant increasing trend at 5500–6000 m (s = 0.001 cm decade$^{-1}$, p = 0.86). The spatial pattern of SCD showing a significant decreasing trend is similar to that of SD. In addition, we found that, regarding SCD, there is a significant decrease in the central Himalayas and the northern edge of the Kunlun Mountains (<−1 d decade$^{-1}$, p < 0.05), and there is a significant increase in the Bayan Har Mountains and southern Kunlun Mountains (>1 d decade$^{-1}$, p < 0.05). For most other areas, there is no significant trend in SD and SCD, especially in northern TP and the Qaidam Basin. The reason for this phenomenon may be that most of these areas are snow-free areas with less snow (Figure 2), resulting in very weak snow changes. The spatial analysis of SD trend variations by Bian et al. [19] showed that MERRA-2 has a significant decreasing trend in the far western TP, JRA-55 has a significant decreasing trend in the western and a small part of the southern TP, while MW has no significant change trend in all areas. With respect to SCD, the areas with a decreasing trend focus on the far western and southern TP for MERRA-2, the western and eastern TP for JRA-55, and the north of the central TP for MW. The above analyses illustrate further the uncertainty of the reanalysis data.

The main causes of snow cover change over TP are discussed from the perspective of climate change. More precisely, the volume of snow cover is related to the temperature threshold (0 °C), which has not changed substantially during the recent global warming [18]. In addition, the terrain of TP is complex, and the climatic conditions vary from region to region [79,80]. Studies have shown that TP warming is accelerating. Except for the Qaidam Basin, the temperature gradually decreases from south to north. Precipitation tends to increase, but the increased rate decreases from southeast to northwest [81,82], thereby showing different degrees of impact on snow. There is a significant negative correlation between snow cover and temperature, but the relationship between snow cover and precipitation is spatially heterogeneous (Figures 6 and 7). Specifically, in the Kunlun Mountains, the valleys of southern Tibet, and the central and eastern areas of TP, there is a negative correlation between snow cover and precipitation; while in the northern TP, the Qaidam Basin, and the southeastern edge of TP, there is a positive correlation between

snow cover and precipitation. The possible reason for this phenomenon is that the high-altitude mountains in the west and north have lower temperatures and more precipitation, which helps to form snow cover. For example, in the Kunlun Mountains and the Himalayas, there is a negative correlation between temperature and SD/SCD. However, the correlation between precipitation and SD/SCD changes greatly at different altitudes. In the Himalayas, precipitation is positively correlated with SD and SCD at the altitudes of 4000–5000 m and >5500 m. In the Kunlun Mountains, there is a positive correlation between precipitation and SD and SCD at an altitude of 4400 m and >6000 m. The central and eastern regions, however, are mostly broad plains at lower altitudes, where higher temperatures discourage the presence of snow and more precipitation facilitates snowmelt [83,84]. Furthermore, the competitive effect of temperature and precipitation leads to differences in snow cover changes in different areas of TP [78]; Bian et al. [19] insist that the relatively small change of the end date of the annual maximum consecutive snow-covered days (CSCDMaxE) is mainly caused by the combined impact of warmer temperatures and more precipitation in March–April.

### 4.2. Analysis on the Causes of Snow Cover Variation at Different Altitudes

The variation range of snow cover at different altitude intervals is significantly different. The research showed that at 1000–5000 m, as the altitude becomes higher, the decrease rate of SD gradually decreases (Table 1), $R_{sd\text{-}p}$ is higher than $R_{sd\text{-}t}$ (Figure 8a), and precipitation turns into snow when the temperature drops below 0 °C, resulting in the positive feedback effect of precipitation that offsets the effect of temperature rise on snow depth [38]. Relatively, the trend of SCD fluctuates greatly (Table 3), where the decrease rate is the largest at 3000–4000 m (s = −0.33, p = 0.22), and $R_{scd\text{-}t}$ is high (Figure 8b). The possible reason is that at lower altitudes, the snow is sparse and shallow, and the temperature is relatively high. Changes in temperature directly lead to ablation or maintenance of snow [18]. Meanwhile, the altitude of 3000–4000 m is particularly sensitive to higher freezing levels. In addition, we found that both SD (0.3 cm decade$^{-1}$, p = 0.47) and SCD (0.4 d decade$^{-1}$, p = 0.90) show insignificant increasing trends when the altitude >6000 m. The reasons are as follows: warming and humidification have become the main characteristics of TP climate change, and temperature is obviously dependent on altitude. [21,26]. At the altitude of >6000 m, precipitation is beneficial to the accumulation of snow cover at low temperatures. This is because at the altitudes of >6000 m, precipitation essentially only falls as snow and this region is not yet sensitive to rising freezing levels. Compared with similar products, the ERA5 dataset has a low resolution (0.25° × 0.25°), which may lead to a less detailed capture of the high mountains. There may be other factors that affect the variation of snow cover at high altitudes, such as valley circulation and sublimation of snow cover caused by strong winds [27]. Therefore, it is necessary to further analyze the influence of other factors on snow cover in the future.

**Table 3.** Long-term change trends of annual SD and SCD from 1979 to 2018 at different altitude intervals.

| Altitude (m) | SD | | | SCD | | |
|---|---|---|---|---|---|---|
| | Slope (cm decade$^{-1}$) | R$^2$ | p-Value | Slope (cm decade$^{-1}$) | R$^2$ | p-Value |
| 1000–2000 | −0.5 | 0.06 | 0.13 | −3.0 | 0.21 | <0.01 |
| 2000–3000 | −0.4 | 0.23 | <0.01 | −2.0 | 0.16 | <0.01 |
| 3000–4000 | −0.2 | 0.03 | 0.29 | 3.3 | 0.14 | 0.02 |
| 4000–5000 | −0.1 | 0.02 | 0.43 | 1.8 | 0.03 | 0.30 |
| 5000–6000 | 0.7 | 0.12 | 0.02 | 2.7 | 0.03 | 0.23 |
| >6000 | 0.3 | 0.01 | 0.47 | 0.4 | 0.01 | 0.90 |
| Overall | 0.04 | 0.01 | 0.81 | −2.3 | 0.06 | 0.10 |

*4.3. Implications and Limitations*

Snowmelt has a significant impact on water resources in the surrounding areas of TP [85]. Zhang et al. [86] propose that snowmelt runoff accounts for 20–30% of the total runoff in major river basins. Li et al. [21] point out that the earlier the snowmelt, the higher the peak of spring runoff. This study found that the snow cover decreases most in spring (Figure 3), which may lead to natural disasters such as droughts and floods in the Yangtze River Basin. It is necessary for relevant departments and residents to take safety precautions [87]. For example, in the autumn and winter of 1997, the temperature was lower and the precipitation was more, which resulted in abundant snow. However, in the spring of 1998, with the increase of temperature, snow decreased rapidly [78], followed by severe floods in The Yangtze River basin of China. Chen [88] believes that it is reasonable to believe that the flood disaster is related to the snow abnormality of TP. As glaciers shrink, permanent snow cover decreases and lake water levels rise [89]. Therefore, monitoring the dynamic changes of snow cover over TP is helpful to provide an important scientific basis for regional water resources management and disaster prevention and control [90].

Remote sensing technology and data assimilation systems provide advanced methods for dynamic monitoring and in-depth study of snow cover. However, due to the unique geographic location of TP, complex climate environment, and insufficient station observational data, the numerical assimilation models in this area generate high temporal and spatial uncertainties [91]. In addition, under the impact of assimilation data, ERA5 shows strong inhomogeneities. The performance test on the ERA5 reanalysis dataset showed that ERA5 has a positive bias of overestimating snow depth. This phenomenon may be caused by excessive snowfall or sublimation of blowing snow [61]. In fact, excessive snowfall over TP is a common bias among climate and forecast models [92]. Wang and Zeng [93] evaluated several reanalysis products in terms of temperature and precipitation, such as ERA-40, ERA-Interim, the MERRA, National Centers for Environmental Prediction/National Center for Atmospheric Research (NCEP/NCAR) reanalysis products, Global Land Data Assimilation System (GLDAS) datasets, and Climate Forecast System Reanalysis (CFSR). The results showed that there is excessive precipitation in other reanalysis data, except for MERRA-2 over TP. Furthermore, ERA5 has a higher horizontal resolution than other reanalysis data, thus allowing a better description of precipitation events. However, the precipitation season cycle in the southeast of TP is mainly summer monsoon precipitation, which reaches 70–90% of the annual precipitation in the region. In addition, when the altitude of this area is above 5000 m, the most snow is acquired in summer. The glaciers in the southern region are considered summer accumulation because most of their snow accumulation occurs during summer monsoon. However, we focus instead on the cold season when the precipitation is less and more falls in the form of snow. When studying the correlation between snow cover change and temperature and precipitation, there is still some uncertainty, and multiple datasets can be compared and analyzed at the same time to reduce the errors brought by a single product [19]. Therefore, to more accurately monitor the dynamic changes of snow cover in TP in the future, it is still necessary to further improve the high-altitude climate monitoring system and require more station SD observations [94]. Meanwhile, the implementation of snow processes requires more accurate snow data for assimilation or simulation studies, which can not only validate data assimilation products, model simulations, and reanalysis datasets, but can also be used for further hydrology, weather, and climate studies [41].

## 5. Conclusions

This paper studies the temporal and spatial characteristics of SD and SCD over TP from 1979 to 2018 using the ERA5 reanalysis dataset. The results show that: (1) the areas with high snow cover were mainly distributed in the Karakoram Mountains, Nyainqentanglha Mountains, the southern edge of the Himalayas, and the Qilian Mountains; (2) in the snow cover year, 83% of the areas experienced a decline in SD and 81% of the areas

experienced a decline in SCD; (3) SD had a sudden change in 1984 and 2015, showing an "up-down-up" trend, and SCD had a sudden change in 2000 m, showing an "up-down" trend; (4) there is a significant negative correlation between snow cover and temperature, while the relationship between snow cover and precipitation has spatial heterogeneity, indicating that snow cover and precipitation have a significant positive correlation in the western and northern regions, and a significant negative correlation in the central and eastern regions; and (5) at different altitudes, the correlation between SD and temperature/precipitation is different from the correlation between SCD and temperature/precipitation. When the altitude exceeds 6000 m, the correlation between SD/SCD and temperature/precipitation becomes not obvious, which may be associated with a variety of factors, and the relevant content needs further study.

**Supplementary Materials:** The following are available online at www.mdpi.com/2073-4441/13/3/307/s1, Figure S1: Interannual variability of SD (black dotted line), SCD (yellow dotted line), temperature (red line) and precipitation (bule line) over TP in autumn, winter, spring from 1979 to 2018, Figure S2. Interannual variability in SD and SCD at various altitude intervals across TP in autumn during 1979−2018, Figure S3. Interannual variability in the SD and SCD at various altitude intervals across TP in winter during 1979−2018, Figure S4. Interannual variability in the SD and SCD at various altitude intervals across TP in spring during 1979−2018, Figure S5. Interannual variability in the SD and SCD at various altitude intervals during 1979−2018 over the Himalayas, Figure S6. Interannual variability in the SD and SCD at various altitude intervals during 1979−2018 over the Kunlun Mountains, Figure S7. The correlation between SD (a)/SCD (b) and temperature/precipitation at different altitude intervals over the Himalayas. The red line represents the correlation coefficient between SD/SCD and temperature, and the blue line represents the correlation coefficient between SD/SCD and precipitation, Figure S8. The correlation between SD (a)/SCD (b) and temperature/precipitation at different altitude intervals over the Kunlun Mountains. The red line represents the correlation coefficient between SD/SCD and temperature, and the blue line represents the correlation coefficient between SD/SCD and precipitation, Table S1. Summary of changing trends in snow cover (i.e., SD and SCD) and environmental variables (i.e., precipitation and temperature) from 1979 to 2018 over TP. The unit of Slope regarding SCD is d decade$^{-1}$, the unit of Slope regarding SD is cm decade$^{-1}$, the unit of Slope regarding Temperature is °C decade$^{-1}$, the unit of Slope regarding Precipitation is mm decade$^{-1}$, Table S2. Long−term change trends of annual SD and SCD from 1979 to 2018 at different altitude intervals in autumn, Table S3. Long−term change trends of annual SD and SCD from 1979 to 2018 at different altitude intervals in winter, Table S4. Long−term change trends of annual SD and SCD from 1979 to 2018 at different altitude intervals in spring, Table S5. Long−term change trends of annual SD and SCD from 1979 to 2018 at different altitude intervals over the Himalayas, Table S6. Long−term change trends of annual SD and SCD from 1979 to 2018 at different altitude intervals over the Kunlun Mountains.

**Author Contributions:** C.Z., N.M. and F.L. conceived and designed the research question. C.Z., N.M. and J.N. constructed the models and analyzed the optimal solutions. C.Z. wrote the paper. C.Z., J.N. and L.Z. reviewed and edited the original and revised manuscript. C.Z., N.M. and F.L. completed revised manuscript. All authors have read and agreed to the published version of the manuscript.

**Funding:** This paper was supported by the Open Fund of Key Laboratory for Synergistic Prevention of Water and Soil Environmental Pollution with grant No. KLSPWSEP-A09.

**Institutional Review Board Statement:** Not applicable.

**Informed Consent Statement:** Not applicable.

**Data Availability Statement:** The data presented in this study are available on request from the corresponding author.

**Acknowledgments:** The authors are grateful to the editors and anonymous reviewers who provided valuable comments and suggestions to significantly improve the quality of the paper.

**Conflicts of Interest:** The authors declare no conflict of interest. The funders had no role in the design of the study; in the collection, analyses, or interpretation of data; in the writing of the manuscript, or in the decision to publish the results.

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
