# Peer review of "Spatio-Temporal Variation Characteristics of Snow Depth and Snow Cover Days over the Tibetan Plateau"

_water, doi:10.3390/w13030307_

Round 1
Reviewer 1 Report
This is well done and important study. There are no objections to the methodology both in the part concerning the snow cover methods, statistics as well as the use of GIS tools and the ERA5 database.
The authors analyzed a multi-year data series that shows the features of snow cover changes taking into account altitude zones over a large continental area. The most interesting thread is conclusion about positive feedback of precipitation, not significant changes of snow depth and variability at various altitude intervals, which demonstrate the disappearance of the trend with the altitude.
The results obtained are important for understanding the regional effects of global warming.
Congratulations :)
Reviewer 2 Report
Zhang et al (2020) examine Snow depth and snow cover days across the Tibetan Plateau (TP) using a reanalysis data set. Having read and reviewed several papers on this topic in recent years, this paper does not advance our knowledge as currently written. Key issues are: 1) that the positive bias for SD of the reanalysis product is not considered. 2) that there is a consistent and significant decline in SCD identified in several other recent studies for most or much of the TP, particularly in a later onset. This highlights the need for this paper to be more focussed on specific regions and elevation bands where significant trends can be found versus averaging so widely in time and space that no significance is found.
24: Indicate area of Tibetan Plateau.
31: “..Intergovernmental Panel on Climate Change [10] reported that between 1951 and 2005, the average temperature in China increased by 1.3 C, which has had an adverse impact on snow cover [11, 12].”
34: Doesnthe SD that Qin and Liu [13} report represent an average value through the winter season or an average maximum?
44: What does “degree” here refer to?
68: is this a specific reanalysis data set you are referring to or in general? If specific name the product.
86: Incorrect area of 200 km2. Would be useful to provide more than a mean elevation or range, do you have a hyspometry to report?
94: “Snowfall is the surface characteristic of TP with the greatest seasonal variations. It is closely related to the East Asian and South Asian monsoons and impacts both drought and floods including on the middle and lower reaches of the Yangtze River [51-53].
111: ERA5 has been shown to capture the spatial distribution of snowcover well, but has been shown to have a signficant positive bias for snow depth. This bias needs to be discussed as it pertains to the SD findings in this paper, note Orsolini et al (2019).
126: That most snowfall occurs in winter is true in terms of area and amount for most of the TP but the high mountains of Asia along the southern part of TP are impacted by the summer Monsoon. Bewteen 70 and 90% of annual precipitation occurs during this time period. The study does not have to evaluate summer but has to acknowledge where it can impact the data set.
171: 7% of the total area of TP?
188: “In each season SCD decreases across ~90% of the TP and increases across ~10% of the TP.”
199: The lack of significant trend in SCD is at odds with the TP work of Bian et al (2020). They note “There was a clear decrease in the annual maximum consecutive snow‐covered days (CSCDMax), and this was characterized by a later begin time and an earlier end time for both MERRA‐2 and JRA‐55, but only a later begin time for MW during the time period 1980–2018.” and “The CSCDMax averaged over the Tibetan Plateau showed decreasing trends in all studied time periods for JRA‐55 and MW and in the last 38 and 10 snow seasons for MERRA‐2 during the time period 1980–2018.”
217: Why is the SCD trend here greater than noted at line 201, 1.62 vs 2.21?
217: Please discuss in more detail the trends lines for SD and SCD and what it means for them to decline at all elevations below 5000 m and increase above that.
228: This is quite an obvious statement. The only important point is what areas tend to not be signficant and is there a pattern there. Also note your elevation SD and SCD graphs also illustrate the temperature dependence as T is the primary variable impacted by elevation.
245: Compare to findings of Bian et al (2020).
278: The result is not consistent with Bian et al (2020). It is also only partly consistnet with Xu et al {75} who report changes consistent with Bian et al (2020) they note declines in autumn “In summer and autumn, snow depth and the number of snow-cover days show a significant decreasing trend for most sites. The duration of snow cover exhibits a significant decreasing trend (−3.5 ± 1.2 days decade−1), which was jointly controlled by a later snow starting time (1.6 ± 0.8 days decade−1) and an earlier snow ending time (−1.9 ± 0.8 days decade−1) consistent with a response to climate change.” Wang et al [18] also note the later start of snow coverage in some area “Later SCS, earlier SCM, and thus decreased SCD mainly occurred in the areas with elevation below 3500 m a.s.l.,”. The later onset of snowcover is a conistent result for the TP from these three studies, how does that fit your results?
280: The opportunity for this paper to have value is to look at specific regions and identify those areas with significant trends that do not exist for the entire TP.
Bian, Q., Xu, Z., Zheng, H., Li, K., Liang, J., Fei, W., et al. (2020). Multiscale changes in snow over the Tibetan Plateau during 1980–2018 represented by reanalysis data sets and satellite observations. Journal of Geophysical Research: Atmospheres, 125, e2019JD031914. https://doi.org/10.1029/2019JD031914
Orsolini, Y., Wegmann, M., Dutra, E., Liu, B., Balsamo, G., Yang, K., de Rosnay, P., Zhu, C., Wang, W., Senan, R., and Arduini, G.: Evaluation of snow depth and snow cover over the Tibetan Plateau in global reanalyses using in situ and satellite remote sensing observations, The Cryosphere, 13, 2221–2239, https://doi.org/10.5194/tc-13-2221-2019, 2019.
Reviewer 3 Report
see attached document

Round 2
Reviewer 2 Report
Zhang et al (2020) examine Snow depth and snow cover days across the Tibetan Plateau (TP) using a reanalysis data set. The authors have expanded the discussion on trends in their data regionally and comparison with other studies. As noted before looking at SD and SCD for the entire TP has been completed several times. By focusing primarily on this data set that encompasses all of the TP, ranges that have different responses and are the most important/significant changes are mostly lost or poorly quantified. This data is in hand, this paper can be of more than average value if more analysis of a couple of sub-regions is compelted. I am not suggesting the authors focus on every sub-region or on every aspect, but instead pick two that illustrate the differing response. This is particularly true for Figure 5 and 8.
202: A bit more regional discussion of the results of Figure 3 is warranted. It is striking in the extent of the change in SCD along the southern edge of TP, the Himalaya. This is an area where the rise in freezing line has been noted (Baker Perry, et al 2020). The SE TP has the most striking loss of snow with both SD and SCD declines in Figure 3. Kunlun also has a seasonally the most striking change in SCD in winter to the positive.
226: By looking at the elevation bands across all of TP the most important trends will be lost. I would suggest providing this analysis by elevation band for two different areas, not all of the sub-regions the Kunlun and the Himalaya. The results indicate more significant trends due to temperature impacts in the Himalaya.
274: Again here the SD and SCD comparison at a particularl elevation band could be done for the same two sub-regions of TP Himalaya and Kunlun.
310: Analysis suggested for the two sub-regiions in previous comments could better address this insight.
340: The suggested analysis of the two sub-regions would help better define this as well.
353: The elevation zone from 3000-4000 m is particularly sensitive to higher freezing levels during the winter.
363; Above 6000 m precipitation essentially only falls as snow and this region is not yet sensitive to rising freezing levels.
396; Note that your highest elevation bands in the south do get most snow in summer. The glaciers in the souther region are considered summer accumulation because most of their snow accumulation occurs during summer monsoon. This is in areas above 5000 m, below that it is rain.
https://www.cell.com/one-earth/fulltext/S2590-3322(20)30541-8#%20
Reviewer 3 Report
see attached file
